# Multiply Recurrent Composite Hemangioendothelioma of Penis with Histologic Progression to High-Grade Features

**Chau M. Bui \* and Bonnie Balzer**

Department of Pathology and Laboratory Medicine, Cedars Sinai Medical Center, Los Angeles, CA 90048, USA
\* Correspondence: chau.bui@cshs.org

**Abstract:** Composite hemangioendothelioma (CHE) is a very rare low-grade malignant vascular neoplasm. Here, we present the first case of it occurring on a penis with two local recurrences over a 9 year span and its progression to a high-grade morphology.

**Keywords:** composite; hemangioendothelioma; retiform; angiosarcoma

## 1. Introduction

Composite hemangioendothelioma (CHE) was first described in the literature by Nayler et al. in 2000 [1] as a rare vascular tumor of low-grade malignancy with a tendency for local recurrence. The neoplasm exhibits multiple components representing various subtypes of hemangioendothelioma, including retiform hemangioendothelioma (RHE), spindle cell hemangioendothelioma (SCHE), Kaposi-form hemangioendothelioma (KHE), epithelioid hemangioendothelioma (EHE), well-differentiated angiosarcoma (AS), and other types of benign hemangioma [1]. It occurs predominantly in females and in adults, with a mean age of 42.5 years [2]. CHE is locally aggressive and rarely metastasizes. CHE is not well described in the literature. Here, we report the first case arising in the penis with local recurrences and its progression to high-grade features.

## 2. Case Presentation

A man in his 40s presented with a scar on the left surface of the glans penis measuring $1.8 \times 1.5 \times 1.0$ cm$^3$. There was an area of indurated skin lateral to the scar. He underwent a partial penectomy with a reconstruction of the glans penis. The penile lesion was completely excised. The skin was dissected free from the underlying cavernous tissue with clear margins grossly. Histologically, the lesion was relatively poorly circumscribed, nonencapsulated, and it had a lobulated appearance in the center, with an admixture of vascular retiform and epithelioid patterns in a fibrotic background. Intravascular proliferation was seen in some of the vascular structures. The retiform structures were composed of long, branching, thin-walled vessels with a single layer of bland hobnail endothelial cells. The epithelioid component showed crowded epithelioid cells with vesicular chromatin and inconspicuous nucleoli. There were scattered extravasated erythrocytes in some areas. Cytologic atypia was not significant, but the mitotic figures were easily found. Necrosis was completely absent (Figure 1). The overlying skin showed unremarkable epidermis and superficial lymphangiectasia. Notably, the proliferation of the atypical epithelioid cells was present at the deep proximal right margin (the 3–6 o'clock margin). The tumor cells expressed a strong positivity for vascular markers, including CD31, CD34, and FLI-1, whereas cytokeratin AE1/AE3, HHV-8, and SMA were found to be negative.

Six months after the excision, the patient noticed a slowly enlarging, painless lesion at the surgical site. He did not return for a follow-up until seven years after the initial resection. Magnetic resonance imaging (MRI) was performed, and there was no evidence of metastasis. The lesion primarily involved approximately a quarter of the glans of the penis at the 3–6 o'clock position. The tumor did not grossly involve the urethra or the shaft of the

penis proximal to the glans penis. A partial penectomy was performed to remove all of the grossly palpable tissue with the goal of achieving completely negative margins. The patient did not receive any other treatment. Upon gross examination, the specimen consisted of a $2.8 \times 2.1 \times 1.0 \text{ cm}^3$ portion of the glans penis. The sectioning showed a $1.3 \times 1.0 \times 0.8 \text{ cm}^3$ well-demarcated, pink-tan, firm, hemorrhagic, ulcerated lesion that abutted the resection margin. The margins were involved at the periphery and base grossly. Microscopically, the tumor retained features of retiform hemangioendothelioma and displayed a notable increase in cellularity and cytologic atypia compared to that of the primary. There were up to six mitotic figures per ten high power fields (six MFs/ten HPF). However, no overt tumor necrosis was identified (Figure 1). The tumor was present at the peripheral and deep margins. The immunophenotype was similar to that of the primary, diffusely, and strongly positive for CD31, CD34, ERG, and FLI-1 and negative for cMYC, HHV8, and AE1/AE3 (Figure 2). A next-generation sequencing (NGS) panel with 58 gene fusions (Cleveland Clinic Foundation panel) was performed, and no gene fusion was detected.

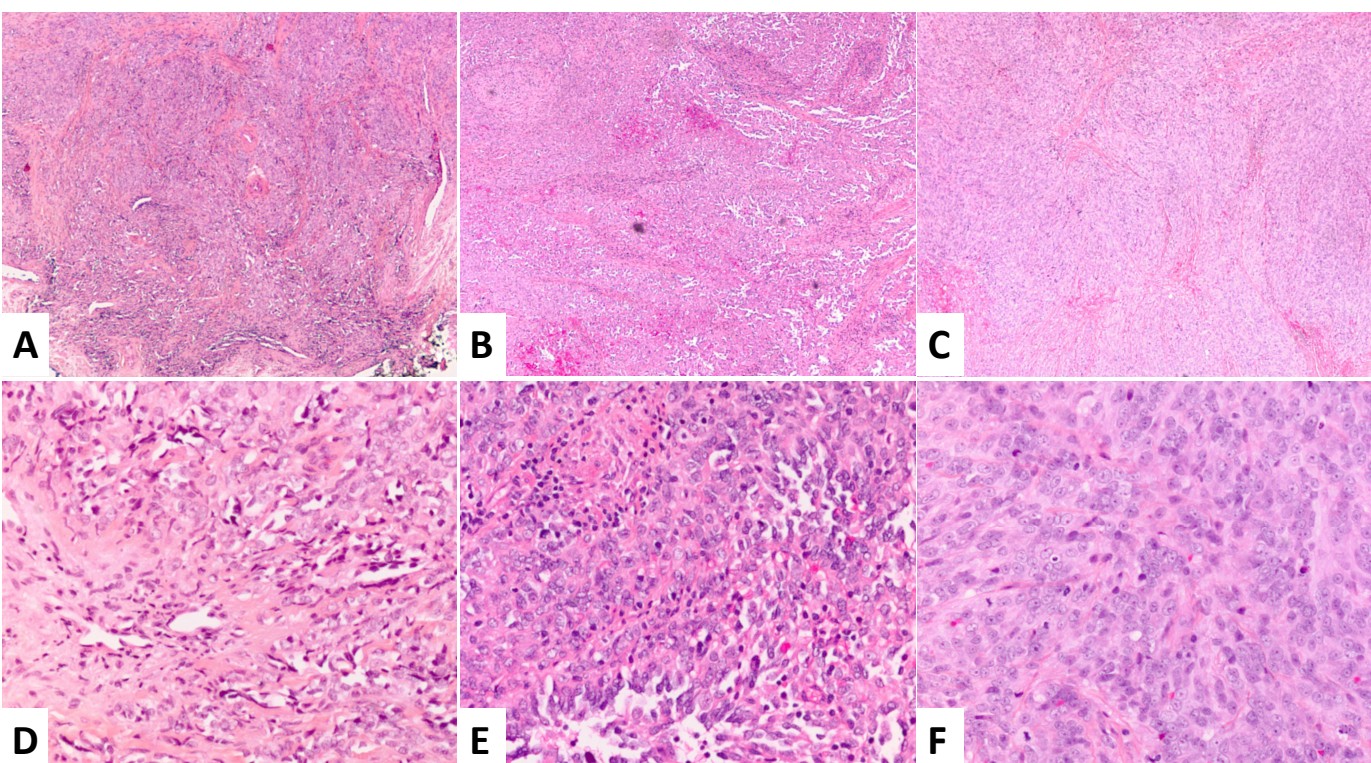

**Figure 1.** Morphologic features of the three lesions at 40× and 200× magnification. (**A**,**D**) Primary lesion in 2008 with admixture of retiform and epithelioid patterns and rare MFs. (**B**,**E**) First recurrent lesion in 2015 with increase in cellularity and cytologic atypia and 8 MFs/10 HPF. (**C**,**F**) Second recurrent lesion in 2017 with pleomorphism, prominent nucleoli, and >10 MFs/10 HPF.

Two years after the second resection, the patient re-presented with a slowly enlarging lesion at the site of his last surgery, which he first noticed six months prior. He complained of intermittent white and bloody discharge at the left lateral subcoronal region after intercourse. No metastasis was identified by imaging. There was an apparent recurrence at the 6 o'clock position, at the junction between the glans and the penile shaft. There was an approximately 4 mm area of skin breakdown and an approximately 1.5 cm indurated region underneath it. The area of induration was mobile, and it was not attached to the underlying corpora. He underwent an excisional penile biopsy to completely remove the indurated lesion and local radiation therapy for six months. The gross examination showed a $2.3 \times 1.8 \times 0.6 \text{ cm}^3$, irregularly shaped portion of tan skin with a firm, white, bosselated lesion raised above the surrounding epidermis of approximately 1.0 cm. The tumor con-

tained areas that were histologically similar to the previous recurrence in addition to a solid component showing non-vasoformative epithelioid cytomorphology and several high-grade features, including an increased nuclear size, prominent nucleoli, brisk mitotic figures (>10 MFs/10 HPF), and increased pleomorphism. The resection margins appeared to be uninvolved. Similar to the prior lesions, the tumor cells showed positive endothelial markers, including ERG, CD31, and FLI-1, however, the tumor cells also expressed patchy synaptophysin positivity and rare MYC positivity. It was negative for TFE3, HHV8, and CAMTA-1. The fluorescence in situ hybridization (FISH) was performed and showed no abnormalities in MYC, FOS, FOSBI, or TFE3.

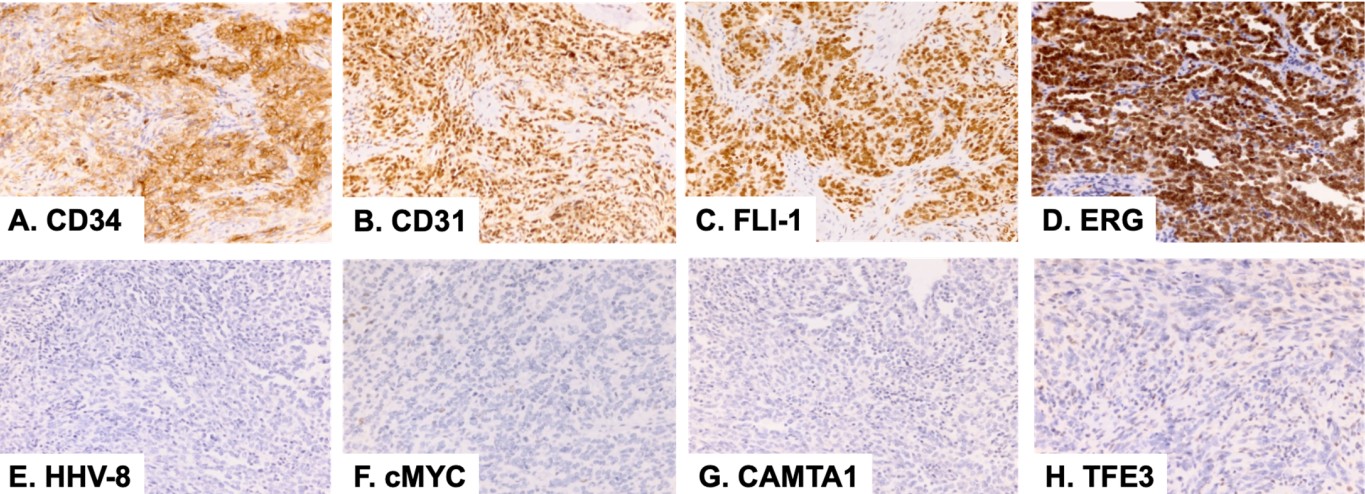

**Figure 2.** Immunohistochemical stains of the second lesion (2015) at 100× magnification.

Since then, the patient has been followed closely for over five years, and no recurrent or metastatic lesions have been observed clinically or radiographically.

## 3. Discussion

CHE can present as a nodule, plaque, or ulcerated tumor, with a size ranging from 0.4 to 30 cm. They are found predominantly in the dermis and subcutis of the distal extremities. The other locations include the head and neck, paraspinal region, gluteal region, kidney, spleen, retroperitoneum, and mediastinum [1,3–9]. This is the first reported case of this entity occurring in the penile region. CHE rarely metastasizes, but it has a high local recurrence rate, with the time to recurrence ranging from 4 months to 10 years after the original excision. Therefore, the patients must be followed closely. The first-line treatment is wide local surgical excision [10].

In this case, the patient had two local recurrences in the setting of the inadequately resected margins. The challenge in obtaining an adequate margin in this case was likely due to multiple factors, including the sensitive location of the tumor on the glans penis and desire to preserve sexual function post-operatively, as well as the nature of CHE, which often has ill-defined borders that are difficult to delineate grossly. Since the last excision, this patient has not had any recurrence after five years of follow-up, which is perhaps due to the negative margins with the most recent excision and use of adjuvant radiation therapy.

CHE is a poorly circumscribed vascular neoplasm with infiltrative margins containing a mixture of at least two morphologically distinct vascular tumor components. A retiform hemangioendothelioma-like pattern is the most common dominant component that is seen [3]. Therefore, RHE is the most common differential of CHE. RHE is a nodular or plaque-like lesion that arises in the skin or subcutis of adults. RHE exhibits distinctive elongated, retiform blood vessels, resembling the rete testis. The vascular spaces are lined by endothelial cells with a hobnail appearance and hyperchromatic nuclei. The evaluation of the entire lesion to look for other vascular growth patterns is a helpful way to differentiate

CHE from RHE. Similar to CHE, RHE shows reactivity with CD31, CD34, FLI-1, EGR, and podoplanin (D2–40) immunohistochemistry, and it does not express GLUT1 and HHV8. RHE also has a very low risk of metastasis, but it has a high rate of local recurrence [10].

The other important differentials include AS and EHE. The patients with CHE undergo a complete resection with a close follow-up. In contrast, the patients with AS and EHE receive radiation and/or chemotherapy in addition to the resection. Differentiating CHE from AS can be challenging as CHE very often contains angiosarcomatous and/or benign angiomatous components [1,10]. Shon et al. reported 23/38 cases of primary cutaneous AS expressing cMYC by IHC and showing a cMYC gene rearrangement by FISH [11]. In contrast, the cMYC gene rearrangement has not been reported in CHE to date. A high mitotic figure count is more commonly seen in AS than it is in the angiosarcomatous component of CHE [1,12,13]. CHE with angiosarcoma-like areas is considered to be more aggressive compared to CHE with no angiosarcomatous features [14].

EHE is a malignant tumor of soft tissue, bone, and solid organs. It is composed of epithelioid cells with abundant, eosinophilic, and vacuolated cytoplasm arranged in cords, nests, or small aggregates. It has a characteristic myxoid to hyaline stromal matrix. Similar to CHE, EHE exhibits infiltrative growth with a low mitotic rate [15]. Most EHE cases (86%) have a characteristic t(1;3)(p36.3;q25) translocation, leading to *WWRT1::CAMTA1* gene fusion [16]. A subset of EHE contains YAP1::TFE3 gene fusion [16]. CAMTA1 immunohistochemistry can also be used as a marker to support the diagnosis of EHE. In contrast, CHE is negative for CAMTA1. *YAP1::MAML2*, *EPC1::PCH2*, and *PTBP1::MAML2* gene fusions have been reported in a few cases of CHE [17–19].

To our knowledge, this is the first reported case of CHE involving genital skin in the literature to date. In 2017, Perry et al. reported 11/11 cases of CHE expressing synaptophysin and displaying more aggressive behavior than that which is typically described in other cases of CHE [19]. Distant metastases of the bone, brain, liver, and lung were reported in half of these 11 patients [19]. The second recurrent lesion in this case also showed patchy positivity with synaptophysin IHC. This case highlights the potential for these tumors to progress and develop high-grade features. Although the tumor in this case exhibited a fairly indolent course without metastasis, it is interesting that the delay between recurrences was significantly shorter after the emergence of high-grade features. The clinical behavior of CHE remains incompletely defined because of their rarity. However, as reported cases of CHE increase in number, it may become possible to correlate their histologic, immunohistochemical, and genetic features to predict which tumors may be biologically aggressive.

**Author Contributions:** Conceptualization, C.M.B. and B.B.; resources, C.M.B.; data curation, C.M.B.; writing—original draft preparation, C.M.B.; writing—review and editing, C.M.B. and B.B.; visualization, C.M.B.; supervision, B.B. All authors have read and agreed to the published version of the manuscript.

**Funding:** This research received no external funding.

**Institutional Review Board Statement:** Not applicable.

**Informed Consent Statement:** Written informed consent has been obtained from the patient to publish this paper.

**Conflicts of Interest:** The authors declare no conflict of interest.

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
