# Peer review of "Multiply Recurrent Composite Hemangioendothelioma of Penis with Histologic Progression to High-Grade Features"

_dermatopathology, doi:10.3390/dermatopathology10010005_

Round 1

Reviewer 1 Report

1. Please comment on the possible modes of regrowth of this tumor- inadequately sampled margin, local metastasis?

2. The microscopic images as displayed are too small. I suggest that they be printed larger; if not, the retiform structures should be shot at higher magnification

Author Response

Dear reviewer,

Thank you for reviewing our manuscript! I have added to the discussion a comment about factors likely contributing to the regrowth of this tumor in this case. I have also made the microscopic images larger - I hope that the retiform structures are more easily discernible now.

Kind regards,
Chau Bui

Reviewer 2 Report

The case presentation must be improved because the consecutio tempoum of the events is not completely clear.We need a clinical picture to know in a better way the morphology of the nodule or plaque and for a better clinical pathological correlation.

Author Response

Dear reviewer,

Thank you so much for spending time reviewing our manuscript! I have made changes as you suggested. I rephrased the clinical history to make the timing/sequence of events more clear. I have also included a more detailed description of the clinical morphology of the lesion. Unfortunately, the patient did not consent to having photographs of the lesion published and did not provide any such images to the clinician to be uploaded into the patient chart. 

Kind regards,

Chau Bui